# Cost Model for Biogas and Biomethane Production in Anaerobic Digestion and Upgrading. Case Study: Castile and Leon

**DOI:** 10.3390/ma16010359

**Published:** 2022-12-30

**Authors:** Laura Sánchez-Martín, Marcelo Ortega Romero, Bernardo Llamas, María del Carmen Suárez Rodríguez, Pedro Mora

**Affiliations:** Universidad Politécnica de Madrid, ETSI Minas y Energía, C/Alenza 4, 28003 Madrid, Spain

**Keywords:** climate change, upgrading, biogas production, biomethane production, CO_2_ emissions

## Abstract

The increase in pig production is a key factor in the fight against climate change. The main problem is the amount of slurry which causes environmental problems, therefore optimal management is needed. This management consists of an anaerobic digestion process in which biogas is produced and a subsequent upgrading process produces biomethane. In this study, a comparison of different biomethane production systems is completed in order to determine the optimum for each pig farm, determining that conventional upgrading systems can be used on farms with more than 11,000 pigs and, for smaller numbers of pigs, the biological upgrading system. The implementation of these technologies contributes to reducing fossil energy demand and greenhouse gas emissions by using biogas and biomethane as heat, electricity or vehicle fuel.

## 1. Introduction

Over the last two decades, the representation of renewable energy in the global power generation matrix has experienced a significant growth [1]. According to the Energy Information Administration (EIA) [2], production of electricity from renewable energy came close to 451,000 TWh in 2012, and 59% of the global net addition of power generation capacity came from renewable energy in 2014 [3,4]. Renewable energy grew by 7.6% with an aggregate capacity of 176 gigawatts in 2019 [5] and global renewable energy capacity increased by more than 260 gigawatts in 2020, nearly 50% above the growth recorded in 2019 [6].

China is the world leader in renewable energy generation accounting for 21% of the global share of renewable energy, while the United States provides about 11% of the global share [7,8]. In other words, it is 9.95 × 10^−3^ TJ/inhabitant in China and 22.9 TJ/inhabitant in United States, both data that were obtained from 2019 [2]. Brazil is also in third position with 10%, but aims to increase its share to around 19% in 2020 [9,10]. On the other hand, Spain was 11th in the world rankings in renewable electricity generation capacity in 2018 [11]. It also had an energy consumption from renewable energies of 7.4% in 2019 [12] while it was 21.22% in 2021 [13], having a carbon footprint of 0.19 tCO_2_eq/MWh in 2019 and 0.14 tCO_2_eq/MWh in 2021 [14].

Reducing greenhouse gas emissions and the use of fossil fuels are great challenges facing our society [15,16]. The European Commission’s proposal to reduce greenhouse gas emissions by at least 55% by 2030 compared to 1990 puts Europe on a responsible path to reach climate neutrality by 2050 [17].

For an energy source to count towards the renewable energy targets it must meet the sustainability criteria, where the European Union aims for the increased use of renewable sources or renewable energies that contribute to the reduction in greenhouse gas emissions. The criteria specify the minimum required CO_2_eq emissions-savings compared to a standardized fossil fuel [18]; these minimum required savings are set to increase from 50% for installations producing liquid and gaseous biofuels for use in transportation to 70% from 2021 [19].

The recent uprise in organic waste generation has necessitated the designation of organic waste by governments worldwide (e.g., food waste (FW) and cow slurry (CS)) as a high-priority waste stream because of its significant socio-economic and environmental implications [20]. One of the main solutions proposed is the conversion of this waste into fuel. One method of converting these organic waste materials into useful fuel is through anaerobic digestion. The biogas produced can be used to power internal combustion engines or small gas turbines, burnt directly for cooking, or for space and water heating [21]. In addition, a digestate is produced in this process which can be used as a biofertilizer [22].

Anaerobic digestion can be applied to several types of waste: from the semi-solid to liquid and gaseous waste [23,24]; this process has long been applied to the treatment of domestic and industrial wastes/wastewaters [25]. Furthermore, the use of agricultural material such as manure, slurry and other animal and organic waste for biogas production has, in view of the high potential to reduce greenhouse gas emissions, significant environmental advantages in terms of heat and power production and its use as biofuel [26], e.g., in transport. In the same way, the potential for greenhouse gas emission reductions in the transport sector according to the European Green Deal will be 90% by 2050 [27].

The biogas can serve facilities directly, or even the domestic or consumer-end level as a reference to determine the potential and development of the anaerobic process of the organic waste mixture, from the point of view of the performance feedstock [28].

The renewable energy derived from biogas and the biofertilizer is characterized by considerably lower carbon footprints than fossil fuel-powered energy and chemical fertilizers [29]. Compared to fossil fuels in the EU, biogas and biomethane production can lead to a negative carbon footprint in some cases [30], and biofertilizers allow a reduction in nitrogen fertilizers of about 36% [31]. The values for biogas production from manure include the negative emissions from emission reductions achieved through raw manure management. The value considered is equal to −45 gCO_2_eq/MJ for manure used in anaerobic digestion [19]. Meanwhile, for bioliquids used in electricity production the fossil fuel reference value is 183 gCO_2_eq/MJ [19].

One of the major applications of biogas after upgrading to biomethane is as a fuel in the transport sector and for the construction industry [32]. The conversion of biogas to biomethane can be carried out by means of various technologies, the main differences being in their economic efficiency depending on the quantities to be converted such as: pressure swing adsorption (PSA); water scrubbing; chemical scrubbing; membrane separation and cryogenic separation [33]. Each of these technologies produces biomethane with different purities, so that the PSA technology can achieve a biomethane purity up to 99% [34], water scrubbing can reach more than 97% [35], chemical scrubbing exceeds 98% [36], membrane separation around 98% [37] and cryogenic separation more than 97% [38].

There are also other upgrading systems, such as biological upgrading with microalgae cultivation. Microalgae technology is a promising resource that can simultaneously alleviate environmental and energy issues because it can be used to capture CO_2_ [39] while producing biomass [40], which can be further converted into biofuels or value-added products [41]. In addition, compared to other energy crops, microalgae have several prominent advantages, such as a higher photosynthetic efficiency, lower consumption of water and no requirement for arable lands [42].

Ultimately, all of the above contribute greatly to the circular economy concept with the interconnection of organic waste generation, combined heat and power generation, bio-derived natural gas production (e.g., biomethane) and green fertilizer production [43,44]. And the Figure 1 and Figure 2 show biogas and biomethane production plants in Europe.

Given the importance of having a sustainable system applying the concept of a circular and profitable economy, the aim of this article is the detailed study of different biogas upgrading technologies to biomethane, in order to define the conditions in which the technologies are more profitable and efficient for the generation of electricity in industries or cities with high demand capacity.

## 2. Materials and Methods

### 2.1. Livestock and Biofuel Potential

Spain is at record figures with more than 56.4 million pigs slaughtered per year and around 5 million tons of meat produced. Spain remains in fourth place in the world with these figures, only behind China, United States and very close to Germany in terms of meat production, but surpassing it in terms of the number of animals slaughtered [46]. The annual per capita consumption of pig meat was 30.29 kg in 2017 in China [47], Germany consumed 35.8 kg in 2017 [48], and Spain 54.4 kg in the same year [46], and the annual per capita consumption was 50 kg in 2020 [49]. Moreover, knowing the pig population in 2020, which was 32.6 million head of pigs in Spain and 26 million head of pigs in Germany [46], compared to China, which is 406.5 million [50], and knowing the surface area of each country, the number of pigs per km^2^ is 64.4 in Spain, 72.7 in Germany and 42.4 pigs per km^2^ in China.

The above figures show the great potential of pigs not only in Spain but also in several EU countries. There are three autonomous communities with the highest production in absolute value in Spain: Catalonia, Aragon and Castile and Leon. Among the autonomous communities with the highest production is Castile and Leon, which is the third largest producer of this type of livestock [51]. With the latest update of the pig census provided by the Junta de Castilla y León, for the year 2021 [52], there were a total of 5,269,473 head of pigs in the whole of the autonomous community. A detailed study of the pig population in each of the provinces shows that it follows the sale behavior as reflected in Reparaz et al. [53], i.e., the provinces with the highest pig population are: Segovia, Soria, Salamanca, Zamora and Burgos. For this reason, the research study detailed here focuses on these provinces.

In accordance with the provisions of the Royal Legislative Decree 1/2016, of 16 December, which approved the revised text of the Law on Integrated Pollution Prevention and Control, in its article 2 where this law applies to the industrial facilities that exceed the established capacity thresholds [54], a second selection of pigs was completed within the named provinces. This law aims to reduce and control air, water and soil pollution by establishing a system of integrated pollution prevention and control, the purpose of which is to achieve a high level of protection of the environment as a whole.

Thus, those installations for the intensive rearing of poultry or pigs were selected from those which have more than: 2000 places for fattening pigs over 30 kg;750 places for breeding sows.

After this classification, Table 1 shows the number of farms, their range of pigs on the farms and the pig census of the total number of selected farms in each province.

Furthermore, in accordance with Royal Decree 306/2020, of 11 February, which established basic rules for the management of intensive pig farms and modified the basic rules for the management of extensive pig farms [55], pig farms were classified according to their production capacity, which were expressed in UGM (Livestock Unit) Spanish acronyms, in accordance with the equivalences established for each type of livestock as follows:Small farm: farms with a maximum capacity of 5.1 UGM;Group 1: holdings with a capacity of up to 120 UGM;Group 2: holdings with a capacity exceeding 120 UGM and up to 480 UGM;Group 3: holdings with a capacity exceeding 480 UGM and up to 720 UGM.

With the equivalence in UGM being 0.14 for fattening pigs and 0.30 for breeding sows [56], the number of farms in each group is shown in Table 2, with other farms being those with more than 720 UGM.

With the selection described above, Figure 3 shows the estimated biogas production in each of the provinces under study and taking into account only the selected farms, bearing in mind that they do not represent the total production capacity in each province. With this selection, the order of the provinces from highest to lowest pig population, and therefore from highest to lowest biogas production capacity, is as follows: Soria, Salamanca, Segovia and Burgos y Zamora.

### 2.2. Biogas and Biomethane Production Systems

This research work is based on data from the LIFE SMART AgroMobility project, in which biogas is produced and refined for the production of biomethane. This pilot plant is located in the village of Sauquillo de Boñices in the province of Soria, and the farm has 3450 head of fattening pigs with a manure production of 2.5 m^3^/year per animal [57] which makes a farm manure production of 23.6 m^3^/day. Based on the production capacity of this project, the aim is to quantify production in the 5 provinces selected in the previous section, with the aim of finding out the technically and economically optimal biomethane production process, comparing the aforementioned project with other biogas refining systems.

The slurry generated on the farm was treated for the production of biogas. This slurry was treated in a biodigester where two products were obtained: biogas and biofertilizer. The biodigester was a geotextile in which an operating temperature of 35 °C was used. For this purpose and due to the meteorological conditions of the area where the farm is located, it was necessary to maintain this temperature with a boiler that heated the contents of the biodigester with part of the biogas produced. Regarding the second product, the biofertilizer, one part was used as fertilizer on the crops around the farm and the other part was used in the biomethane production process.

Once the biogas was obtained, an upgrading process was carried out in which biomethane was obtained as a product; this process was nothing more than a cleaning/purification process in the biogas. In this project, a biological upgrading process was chosen, this process was constituted by an absorption column and an open air microalgae culture called raceway, allowing the obtaining of a biomethane purity of approximately 92%. 

The biogas leaving the biodigester was fed into the absorption column which contained archaea and microalgae that acquired the CO_2_ contained in the biogas. In addition, algae were recirculated from the microalgae culture to the absorption column to facilitate the biogas purification process and achieve high concentrations of biomethane. Part of the biofertilizer was also fed into the microalgae culture, which, mixed with the microalgae, produced a biofertilizer with better added value than that obtained from the biodigester.

This model of biogas purification helps to mitigate both environmental and energy problems, due to the capture of the CO_2_ by microalgae and because microalgae produce biomass, which can be converted into high value-added products. In addition, microalgae provide high photosynthetic efficiency, lower water consumption and do not require land for cultivation. It should also be noted that this biogas purification process is modular in construction according to the amount of raw material, in this case, pig slurry.

Both the biogas and biomethane production processes are shown in Figure 4.

The starting data were the characteristics of the slurry, which were acquired from the samples analyzed in the laboratory, in order to know the production of biofuels from pig slurry on the farm, Table 3.

The biogas yield was 0.5 m^3^/kg VS because half of the biogas produced in the biogester was consumed in the boiler to heat the biodigester.

The rest of the upgrading processes with which the study was going to be carried out to find out if it can be implemented in any of the farms were: PSA, Water Scrubbing and Membrane Separation.

Pressure Swing Adsorption (PSA) technology is pressure-variable adsorption to separate gases according to their physical characteristics. It consists of an adsorption column fed by a high-pressure biogas stream where impurities were removed by means of an adsorbent, such as zeolite or activated carbon that removes CO_2_ and N_2_, but not H_2_S which was removed in a previous stage. After these stages, there was a desorption stage that reduced the pressure to atmospheric pressure or vacuum, where gas separation and biogas purification was achieved.

The Water Scrubbing system is a technology based on the greater solubility of CO_2_ in water with respect to methane and it has 3 stages, a first one of absorption and two of desorption. CO_2_ was dissolved in water in the first stage, and separation took place by adding air at atmospheric pressure in the desorption columns.

Membrane Separation technology is the third technology in comparison, and it is a physicochemical technology based on the separation of CO_2_ and H_2_S mainly through semipermeable membranes. There was another previous stage in which H_2_S, water, siloxanes and particles were separated to avoid clogging of the membranes. There were two systems: one was the gas–gas system which operated at high pressure to facilitate selective operation, and the gas–liquid system which was at atmospheric pressure.

### 2.3. Biogas and Biomethane Production Processes Costs

After explaining the production processes of both biofuels in the case study, a comparison of the costs of both production processes and of the three upgrading processes was completed. A detailed indication of the absolute costs for biogas upgrading was difficult due to the high number of influencing parameters and the diversity of biogas plants. However, a general estimation and a comparison amongst different technologies proved to be feasible [58].

The aim of this comparison was to select the optimal system in terms of cost data and product purity in the techno-economic study. In addition, the biogas production process would always be the same and therefore the costs would be the same, regardless of the type of upgrading selected.

In the LIFE SMART AgroMobility project different tasks of earthmoving, civil works and installation of the equipment required for the production processes of the two biofuels were carried out. These activities have the following production costs, Capital Expenditure (CAPEX), Table 4.

The data in Table 4 give a total cost of 82,375 EUR for biogas production versus 119,805.78 EUR for biomethane production, i.e., 24 EUR per pig unit for biogas production versus 35 EUR per pig unit for biomethane production, these costs being for a farm of 3450 head of fattening pigs.

Table 5 shows the maximum, average and minimum flow rates at which each of the upgrading systems operate with which the upgrading system of the named project is to be compared, together with their respective investment costs for each flow rate.

The economy of scale is clearly observable for biogas upgrading. Cost for an installation with a capacity of 250 m^3^/h are in the range of 25 EUR ct/m^3^ while costs drop below 15 EUR ct/m^3^ for capacities above 2000 m^3^/h [58].

In addition, Table 6 shows the percentage ranges of biomethane purity in each of the upgrading systems.

Table 7 shows the slurry flow rates in the farms in each of the provinces under study in order to select which of the upgrading systems in Table 5 can operate in some of the selected farms.

Thus, with the minimum flow rates in Table 5 and the operating ranges in Table 7, it can be seen that the upgrading of Membrane Separation is the only process that could be used either in Soria or Salamanca or in the farms with the characteristics shown in Table 8. The three upgrading systems were discarded for the remaining farms under study.

With the production flows of each of these farms and with Equation (1), the value of their CAPEX is calculated.
(1)$$CAPEX\;\left(€\right)=Flow\;\left(\frac{Nm^{3}}{h}\right)*Investment\;cost\;\left(\frac{€}{Nm^{3}/h}\right)$$

The annual operational expenditures (OPEX) are also taken into account (Table 9), in which the personnel cost is a plant operator who would be the same operator for the production of biogas and biomethane, and the maintenance cost for the upgrading systems obtained from the LIFE SMART AgroMobility project, and the Membrane Separation system from the study carried out by Patterson et al. [60]. For the maintenance costs of the biogas production, only the maintenance of the boiler is considered, which would be a maintenance contract with an annual overhaul.

An OPEX of 43,500 EUR is obtained with the operating costs from Table 9 for the biogas and biomethane production process of the LIFE SMART AgroMobility project, and 48,500 EUR for each of the farms where the Membrane Separation upgrading system can be used, taking into account the biogas and biomethane production process in the OPEX of these farms.

## 3. Results

### 3.1. Results of Biogas and Biomethane Production

With the data in Table 3 and Equation (2), the amount of biogas production in the case study farm, where the LIFE SMART AgroMobility project is located, is obtained:(2)$$Gross\;biogas\;production\left(\frac{m^{3}}{year}\right)=Volatile\;solids\left(\frac{kg}{year}\right)*Biogas\;yield\left(\frac{m^{3}}{kg\;VS}\right)$$

The gross biogas production is obtained from the volatile solids entering the biodigester in Equation (2), because these are the materials from which the biogas is obtained, and the biogas yield which depends on the composition of the waste, in this case pig slurry. The result is 149,402.08 m^3^/year of raw biogas.

In addition, part of the biogas production is used in the heating system as mentioned in the Materials and Methods section; therefore, this biogas consumption in the boiler is taken into account for the calculation of the net biogas production, leaving a net production of 105,057.43 m^3^/year. Equation (3) shows the same result but with energy units (kWh/year) based on the calorific value of the biogas. The calorific value of biogas is between 6 and 6.5 kWh/m^3^, the average value of 6.3 kWh/m^3^ is chosen [61]:(3)$$Net\;biogas\;production\;\left(\frac{kWh}{year}\right)=Net\;biogas\;production\;\left(\frac{m^{3}}{yer}\right)*Calorific\;value\;of\;biogas\;\left(\frac{kWh}{m^{3}}\right)$$

Therefore, the biogas production is 661,861.78 kWh/year, which would be the same amount for biomethane production, since both biofuels have the same amount of methane, which is the energy input of the biofuel. This production is obtained on the farm where the LIFE SMART AgroMobility project is located, so that the production of each of the provinces under study are shown in Table 10. In addition, the consumption of natural gas in each province [62] is shown, which is the fuel to be replaced.

### 3.2. Technical Feasibility: Biogas vs. Biomethane

The aim of this study is to show the technical feasibility of comparing the production of biogas and biomethane, as well as the comparison between the different production systems.

The production of biogas and biomethane is technically proven to be feasible [21,32]. Firstly, biogas production in the geotextile biodigesters is being achieved in other latitudes where the weather conditions are favorable to maintain the operating temperature at 35 °C. In both the location of the farm under study and in the rest of the provinces, it is easy to maintain the temperature with a boiler.

As for biomethane production, the biological upgrading system has also been tested. Both the absorption column and the microalgae culture promote CO_2_ removal [39] and other trace components, all because the microalgae and bacteria contained in the column and in the culture are able to capture the named components and leave a high concentration of methane. The performance of the Membrane Separation upgrading system, as all other technologies were discarded in Section 2.3 Biogas and biomethane production processes costs, is also proven for the operating flow rates shown in Table 5, as it is a well-established system.

### 3.3. Costs of the Whole Comparative Processes

Based on Section 2.3 Biogas and biomethane production processes costs, a comparison was made to select the optimal system. Firstly, Table 11 shows the costs of producing both biofuels in the provinces under study, based on the production methods selected in the project.

Recalling that, as shown in Section 2.3 Biogas and biomethane production processes costs, there is a cost of 24 EUR per pig unit for the production of biogas and 35 EUR per pig unit for the production of biomethane, for this second reason and as there are three farms where another type of upgrading (Membrane Separation) can be used, the CAPEX costs are calculated in these farms by means of Equation (1) and the costs per pig unit, shown in the Table 12.

These results show that for these three farms and in terms of the CAPEX costs, the most economical choice would be to use the Membrane Separation upgrading system instead of the one used in the project. In order to continue with the choice of the most economical option, the OPEX costs are shown in Table 13.

Making an OPEX cost per pig unit of 3.50 EUR for biogas production and 9 EUR per pig unit for biomethane production. For the three farms where the Membrane Separation system can be used, the OPEX cost is calculated to be 36,500 EUR for each of the farms and the following costs per pig unit:Ágreda: 2.16 EUR/pig;Los Rábanos: 2.04 EUR/pig;Valdunciel: 2.92 EUR/pig.

Noting that the OPEX cost for these three farms is lower using the Membrane Separation upgrading system compared to using biological upgrading, this type of upgrading is selected for these farms.

### 3.4. Revenues from the Entire Comparative Processes

The income from biogas and biomethane production on the farms is calculated based on the investment and operation and maintenance costs. Therefore, the first thing to quantify is the income obtained by replacing natural gas with fuel of renewable origin, since using the biofuel produced on the farms replaces the natural gas of fossil origin. Thus, the current market price is used in order to determine this income, 0.11504 EUR/kWh [63] to 3 April 2022.

Another income is due to the tons of CO_2_ avoided by the replacement of fossil fuel by biofuel, so, in order to calculate the tons of CO_2_ avoided by such replacement in the five chosen provinces, these tons are calculated with Equation (4). The calculation is based on the emission factor for natural gas, which is 0.182 kg CO_2_/kWh [64], and the energy value of biofuel production obtained from Table 10:(4)$$CO_{2}\;avoided\;\left(\frac{t}{year}\right)=Emission\;factor\;\left(\frac{kg\;CO_{2}}{kWh}\right)*Net\;biogas\;production\;\left(\frac{kWh}{year}\right)$$

To calculate the revenue from this avoided CO_2_, the average value of the cost of CO_2_ in 2022 is used, which is 84.79 EUR/tCO_2_ [65]:(5)$$CO_{2}\;revenues\;\left(\frac{€}{year}\right)=CO_{2}\;avoided\;\left(\frac{t}{year}\right)*CO_{2}cost\;\left(\frac{€}{t}\right)$$

Table 14 shows the revenues that would be generated by carrying out the above-mentioned production processes.

This results in a biofuel production of 275,282,895.46 kWh/year, from Equation (4) in the five provinces under study, and avoiding 50,101 t/year of CO_2_. The following total income is obtained: 31,668,544.29 EUR/year for natural gas replacement and 4,248,105.08 EUR/year for avoided CO_2_ emissions.

### 3.5. Feasibility of Installing Upgrading Systems

Taking into account the costs and revenues involved in the whole process under study, the feasibility of biomethane production is then carried out in order to quantify in which livestock facilities the installation of biological upgrading would be viable and in which facilities the Membrane Separation upgrading would be feasible. Therefore, with the minimum flow rate data for Membrane Separation in Table 5 (40 m^3^ N/h), it is recognized that this system is feasible for farms with more than 11,000 head of pigs. Based on this data, the following graph shows the CAPEX and OPEX costs for biological upgrading on farms with less than 11,000 head of pigs and for the Membrane Separation system on farms with more than 11,000 head of pigs, shown in the Figure 5.

In addition, the CAPEX values were calculated with reference to the study by Llamas et al., 2021 [66].

As can be seen in the graph, the biological upgrading shows a practically horizontal trend, so that for farms with up to 11,000 pigs there is not a significant variation in cost. This makes this upgrading system cost-effective for farms of the size mentioned above, with 11,000 pigs. Meanwhile, for the upgrading to Membrane Separation, the trend in costs is more pronounced, and there is a greater difference in costs as the number of pigs on the farms increases.

Although the upgrading to Membrane Separation is possible, the trend towards biological upgrading is more economical and makes a significant difference.

## 4. Conclusions

This research work has been carried out in an autonomous community with a large pig population, and the study was implemented on 27.2% of the total pig population of the whole autonomous community after the selection of the farms. This means that the production of biofuel replaces part of the consumption of fossil fuels, in this case natural gas, which is currently used in industries and biofuel can also be used as a vehicle fuel. Thus, the use of biofuel has a positive impact on the country’s decarbonization efforts.

The biogas production system and the above-mentioned upgrading systems are technically tested as production systems, but the use of any of these systems is not possible because there is not enough production to implement them. The Membrane Separation system was selected for three of the farms after carrying out the technical study and leaving the biological system as the only possible system for the rest of the farms, as it is modular and the size can be built according to the size of the farm; thus, obtaining from all the farms under study a total energy of 275,282,895.46 kWh/year and avoiding 50,101 tons of CO_2_ per year.

In addition, with the implementation of these biofuel production systems on farms, there is also a secondary product which is the biofertilizer. This product is of great relevance as the owners of the farms usually have crops to fertilize with this biofertilizer, or in cases where the owners do not have crops, it would also be feasible to use this biofertilizer as they are locations where cereal crops predominate.

Future lines of research include making biogas production more economical, as up to now the most expensive part of its production has been the earthworks and civil works. These could be replaced by a structure that insulates the biodigester from weather conditions, mainly wind and low temperatures. In addition, lines of research could include the possibility of extrapolating the biomethane production system of the LIFE SMART AgroMobility project, or of creating hubs where there is sufficient flow to use any of the upgrading systems.

## Figures and Tables

**Figure 1 materials-16-00359-f001:**
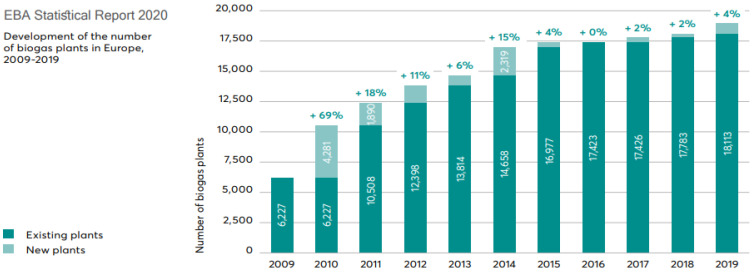
Development of the number of biogas plants in Europe, 2009–2019 [45].

**Figure 2 materials-16-00359-f002:**
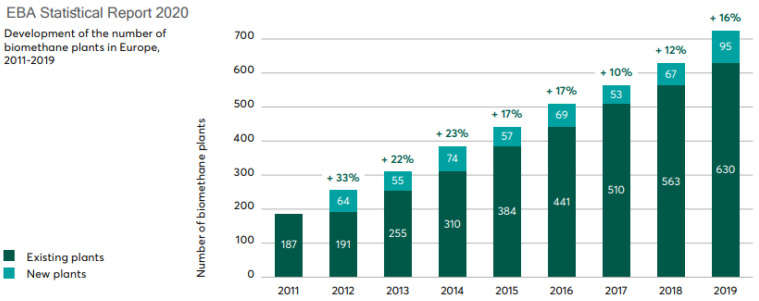
Development of the number of biomethane plants in Europe, 2011–2019 [45].

**Figure 3 materials-16-00359-f003:**
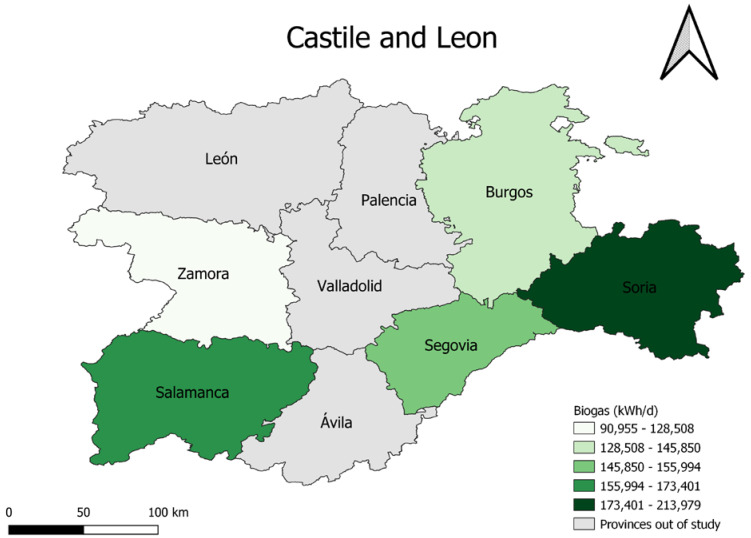
Biogas production of selected farms in Castile y Leon.

**Figure 4 materials-16-00359-f004:**
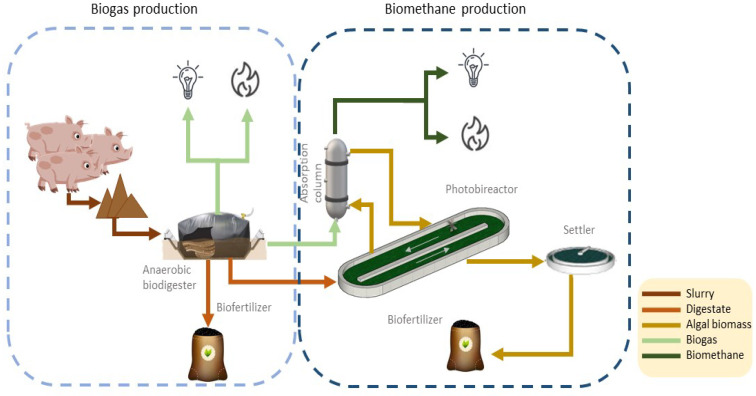
Production systems. Authors’ own elaboration.

**Figure 5 materials-16-00359-f005:**
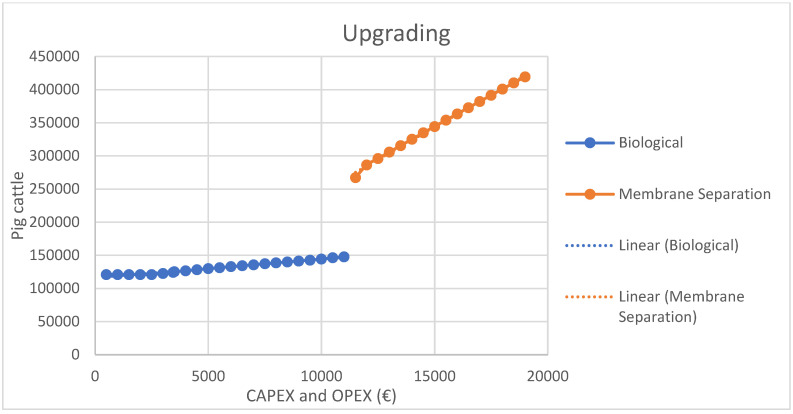
CAPEX and OPEX of upgrading systems.

**Table 1 materials-16-00359-t001:** Pig census and farm characteristics.

Provinces	Number of Farms	Range of Pig Census	Fattening Pigs	Breeding Sows
Segovia	78	1088–7200	268,700	18,879
Soria	93	828–19,890	342,923	47,249
Salamanca	72	1500–12,495	283,554	27,056
Zamora	48	829–6000	169,824	9125
Burgos	69	1000–7580	238,578	29,043

**Table 2 materials-16-00359-t002:** Classification of the farms under study.

Provinces	Group 2	Group 3	Other
Segovia	36	31	11
Soria	45	29	19
Salamanca	28	25	19
Zamora	20	21	7
Burgos	25	33	13

**Table 3 materials-16-00359-t003:** Slurry characteristics. Laboratory data.

Concept	Amount	Unit
Total solids	351,534.30	kg/year
Volatile solids	298,804.16	kg/year
Biogas yield	0.5	m^3^/kg VS

**Table 4 materials-16-00359-t004:** Production costs of biofuels, LIFE SMART AgroMobility.

**Production Costs of Biogas, CAPEX**		
**Equipment**	**EUR**	**EUR/pig**
Earthmoving and civil works	42,400	12.29
Boiler	3880	1.12
Underfloor heating	30,000	8.70
Biodigester	5171	1.50
Slurry recirculation pump	924	0.27
**Production costs of biomethane, CAPEX**		
**Equipment**	**EUR**	**EUR/pig**
Earthmoving and civil works	42,900	12.43
Raceway	27,831	8.07
Absorption column	47,074.87	13.64
Pumps	2000	0.58

**Table 5 materials-16-00359-t005:** Operating ranges and costs of biomethane production on different upgrading [59].

	PSA	Water Scrubbing	Membrane Separation
Maximum flow (Nm^3^/h)	2000	2100	1500
Investment cost (EUR/Nm^3^/h)	1200	1300	1800
Average flow (Nm^3^/h)	1250	1000	800
Investment cost (EUR/Nm^3^/h)	1750	1500	2000
Minimum flow (Nm^3^/h)	500	100	40
Investment cost (EUR/Nm^3^/h)	3000	5100	6000

**Table 6 materials-16-00359-t006:** Biomethane production ranges in each upgrading [59].

	PSA	Water Scrubbing	Membrane Separation
% of biomethane	97–99.5	97–99	98–99.5

**Table 7 materials-16-00359-t007:** Range of production flows in selected provinces.

Provinces	Flow Range (Nm^3^/h)
Soria	2.88–69.14
Salamanca	5.21–43.43
Segovia	3.78–25.03
Burgos	3.48–26.35
Zamora	2.88–20.86

**Table 8 materials-16-00359-t008:** Farms in which Membrane Separation upgrading can be used.

Provinces	Municipalities	Fattening Pigs	Breeding Sows	Flow Rate (Nm^3^/h)
Soria	Ágreda	7310	12,580	69.14
Soria	Los Rábanos	16,102	1832	62.34
Salamanca	Valdunciel	11,655	840	43.43

**Table 9 materials-16-00359-t009:** Operating costs (OPEX).

Concept	EUR	EUR/Pig
Personnel operating the installation	23,000	6.67
Biogas boiler	500	0.14
Prototype maintenance, LIFE SMART AgroMobility	20,000	5.80
Upgrading maintenance, Membrane Separation	25,000	7.25

**Table 10 materials-16-00359-t010:** Biofuel production and natural gas consumption in each province.

Provinces	Biofuel Production (m^3^/Day)	Biofuel Production (kWh/Day)	Natural Gas Consumption (kWh/Day)
Soria	32,551	205,074	4,394,984
Salamanca	25,914	163,257	5,704,860
Segovia	23,992	151,152	3,208,984
Burgos	22,327	140,662	17,688,373
Zamora	14,929	94,056	1,591,329

**Table 11 materials-16-00359-t011:** Biogas and biomethane production costs.

Provinces	CAPEX of Biogas (EUR)	CAPEX of Biomethane (EUR)
Soria	9,316,063.33	12,339,578.62
Salamanca	7,416,376.45	10,352,434.81
Segovia	6,866,469.60	9,986,558.38
Burgos	6,389,939.20	9,293,490.62
Zamora	4,272,731.56	6,214,238.99

**Table 12 materials-16-00359-t012:** Cost of biomethane production with Membrane Separation.

Provinces	Municipalities	CAPEX (EUR)	EUR/Pig
Soria	Ágreda	399,219.03	23.62
Soria	Los Rábano	363,214.89	20.25
Salamanca	Valdunciel	259,438.23	20.76

**Table 13 materials-16-00359-t013:** Biogas and biomethane operating costs.

Provinces	OPEX of Biogas (EUR)	OPEX of Biomethane (EUR)
Soria	1,357,120.00	3,244,390.43
Salamanca	1,080,382.61	2,721,919.57
Segovia	1,000,274.78	2,625,721.30
Burgos	930,855.65	2,443,496.09
Zamora	622,431.30	1,633,882.17

**Table 14 materials-16-00359-t014:** Revenues.

Provinces	Natural Gas Replacement Revenues (EUR/year)	CO_2_ Emission Revenues (EUR/year)
Soria	8,610,991.93	1,155,101.99
Salamanca	6,855,079.82	919,559.14
Segovia	6,346,791.80	851,376.00
Burgos	5,906,324.06	792,290.45
Zamora	3,949,356.68	529,777.50

## Data Availability

Not applicable.

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
