# Peer review of "Cost Model for Biogas and Biomethane Production in Anaerobic Digestion and Upgrading. Case Study: Castile and Leon"

_materials, 2022, doi:10.3390/ma16010359_

Round 1

Reviewer 1 Report

See attached

Reviewer 2 Report

  • This study compared the economic profiles of the biomethane production systems that can help reduce local fossil energy demand and greenhouse gas emissions. The overall wiring, the difference between options. It is recommended for minor revision after addressing some general comments.
  • It would be useful to inform readers about the need for uncertainty analysis that addresses the probability distribution function of variables. The discussion may further the readability. Two practical references that also focus on life-cycle emissions and economic tradeoffs are recommended: 1. Engineering interface between bioenergy recovery and biogas desulfurization: Sustainability interplays of biochar application; 2. Methodological framework for wastewater treatment plants delivering expanded service: Economic tradeoffs and technological decisions. 

Author Response

Thank you very much for your comments. The authors have made changes to the manuscript and your comments have been taken into account. Modifications in the revised manuscript have been included in exchange control.

Reviewer 3 Report

The manuscript entitled Biogas and Biomethane Production Cost Model. Case Study: Castile and Leon shows an interesting study on the cost model of anaerobic digestion from pig slurry.

It will undoubtedly be of interest to the readers of the journal Materials. However, in my opinion, it requires a series of changes, improvements in the presentation of the results and clarifying certain technical aspects before its publication.

That is why I suggest MAJOR REVISION to carry out manuscript improvements and know certain issues of the research and authors’ explanations for some issues.

TITLE

#1 The title must be completed with the objective to be optimized, narrowing the context i.e. it must be included that the cost model of anaerobic digestion of pig slurry and in the pig production system will be studied.

INTRODUCTION

#2 A brief and concise introduction like the one shown is welcome. However, the introduction must contain sufficient material to resolve why it is interesting to study anaerobic co-digestion and its cost model. Emphasis should be placed on the studies that already exist in the literature and why this study stands out the rest, and what differences it entails. Rely on references such as: https://doi.org/10.1016/j.ijhydene.2010.03.135 https://doi.org/10.1016/j.scitotenv.2022.159727 and https://doi.org/10.1016/j.njas.2009.07.006

#3 Does such a study have commercial significance? Can this study find an application area? Scientific explanations should be made on this subject.

MATERIALS AND METHODS

#4 Figure 1: Replace Castilla y León for Castile and Leon, in accordance with the title and the whole manuscript that is written in English

#5 Figure 2: Production systems of... Please complete. Is this figure self-made by the authors? Please indicate.

#6 Table 3: Indicate in the table caption or in the table where the slurry compositional analysis has been taken from (reference is enough)

CONCLUSIONS

#7 The conclusion should be rewritten. Striking results, in which the results are given with more emphasis, should be considered.

GENERAL COMMENTS

#8 Carefully check the writing and English of the manuscript. There are several grammatical errors. I suggest the authors check it out by a native speaker.

#9 Include a list of acronyms.

Reviewer 4 Report

The topic seems to be important and very timely, especially in the context of the fact that biogas and biomethane are most often a by-product of agricultural production and animal farming.

I have a few comments on this article:

-          I recommend at the beginning of the publication to provide a list of abbreviations used in the article so that it is obvious to readers

-          In my opinion, it would be advisable to expand the introductory part with the justification for taking up the topics

-          Since "patent" is not indicated then please remove this point

-          It seems to me that the literature is not adapted to the current requirements - it should be formatted

-          In addition, the share of websites for scientific publications is quite large, perhaps it would be possible to indicate more professional publications, that is, more scientific articles that are based on scientific research conducted

The article is written and thoughtful, with minor changes I recommend its publication

Round 2

Reviewer 3 Report

Authors have successfully addressed all comments.

In my opinion the manuscript is now ready for publication.